# Characterization and Functional Analysis of *OcomOBP7* in *Ophraella communa* Lesage

**DOI:** 10.3390/insects14020190

**Published:** 2023-02-14

**Authors:** Yang Yue, Chao Ma, Yan Zhang, Hong-Song Chen, Jian-Ying Guo, Ting-Hui Liu, Zhong-Shi Zhou

**Affiliations:** 1College of Plant Protection, Hebei Agricultural University, Baoding 071001, China; 2State Key Laboratory for Biology of Plant Diseases and Insect Pests, Institute of Plant Protection, Chinese Academy of Agricultural Sciences, Beijing 100193, China; 3National Nanfan Research Institute, Chinese Academy of Agricultural Sciences, Sanya 572019, China; 4Guangxi Key Laboratory for Biology of Crop Diseases and Insect Pests, Institute of Plant Protection, Guangxi Academy of Agricultural Sciences, Nanning 530007, China

**Keywords:** odorant-binding proteins, plant volatiles, fluorescence binding assay, RNAi, EAG

## Abstract

**Simple Summary:**

*Ophraella communa* Lesage is a specific biological control agent of the invasive weed *Ambrosia artemisiifolia* L. Understanding the molecular mechanism by which *O. communa* recognizes *A. artemisiifolia* will help improve its bio-control effect. Odorant-binding proteins (OBPs) play a vital role in insect olfactory perception. In this study, the sequence and expression characteristics of *OcomOBP7* were analyzed. We obtained the pure protein of *OcomOBP7* by prokaryotic expression and purification, and its binding characteristics were analyzed using a fluorescence competitive binding assay. Finally, we verified the function of *OcomOBP7* in vivo using RNAi combined with an electroantennography (EAG) assay. The results showed that the binding ability of *OcomOBP7* was broad-spectrum and was involved in the host plant localization of *O. communa*.

**Abstract:**

The olfactory system plays a key role in various insect behaviors, and odorant-binding proteins participate in the first step of the olfactory process. *Ophraella communa* Lesage is an oligophagous phytophagous insect that is a specific biological control agent for *Ambrosia artemisiifolia* L. The leaf beetle must identify and locate *A. artemisiifolia* through olfaction; however, its odorant-binding protein (OBP) function has not yet been reported. In this study, *OcomOBP7* was cloned, and its tissue expression profile and binding ability were analyzed using RT-qPCR and fluorescence binding assays, respectively. Sequence analysis demonstrated that *OcomOBP7* belongs to the classical OBP family. The RT-qPCR results showed that *OcomOBP7* was specifically expressed in the antennae, indicating that *OcomOBP7* may be involved in chemical communication. The fluorescence binding assay showed that *OcomOBP7* has an extensive binding ability to alkenes. The electroantennography experiments showed that *O. communa* antennal response to α-pinene and ocimene decreased significantly after interference because the two odors specifically bound to *OcomOBP7*. In summary, α-pinene and ocimene are odorant ligands corresponding to *OcomOBP7*, indicating that *OcomOBP7* is involved in the chemical recognition of *A. artemisiifolia.* Our study lays a theoretical foundation for research into *O. communa* attractants, which is helpful for the better biological control of *A. artemisiifolia* by *O. communa*.

## 1. Introduction

*Ambrosia artemisiifolia* L. is a globally invasive weed that is native to North America. Owing to its extremely high allergenicity and competitive power, *A. artemisiifolia* pollen has caused serious damage to human health and agricultural ecosystems and is listed as a quarantined agricultural weed in China [1,2]. *Ophraella communa* Lesage originated in North America and is an oligophagous phytophagous insect and a specific enemy of *A. artemisiifolia*. Their entire life cycle is accomplished on *A. artemisiifolia* by feeding on the leaves and meristems of *A. artemisiifolia* as both adults and larvae. In recent years, great success has been achieved in the biological control of *A. artemisiifolia* using leaf beetles in southern China [3]. The oligophagous characteristics of *O. communa* make it a perfect model for studying insect–plant communication.

Insects are the most numerous animal group on Earth, 40–50% of which are phytophagous insects. In interspecific communication between herbivorous insects and plants, the sensitive olfactory system plays a crucial role in locating host plants. The olfactory sensation of insects depends on the olfactory sense organ, which is distributed on the antennae and whiskers. The recognition of external odorant molecules requires olfactory proteins, such as odorant-binding proteins (OBPs), chemosensory proteins (CSPs), odorant receptors (ORs), ionotropic receptors (IRs), sensory neuron membrane proteins (SNMPs), and odorant-degrading enzymes (ODEs), for odorant molecule transduction [4,5,6]. During olfactory perception, OBPs first interact with external volatile compounds and transport them to olfactory neurons to activate ORs distributed on the surface of dendritic membranes, which is essential for the normal operation of the insect olfactory system [7].

Insect OBPs are small soluble proteins abundant in the tactile receptors of insects [8,9]. The protein sequences of insect OBPs include highly conserved cysteines with a specific number of amino acid residues between them. For example, classical OBPs include six conserved cysteine residues, of which Coleoptera have two modes: C1-X_23-44_-C2-X_3_-C3-X_36-43_-C4-X_8-12_-C5-X_8_-C6 and C1-X_21-68_-C2-X_3_-C3-X_21-46_-C4-X_8-28_-C5-X_8-9_-C6. The three-dimensional structure of classical OBPs comprises six α-helical domains that form hydrophobic cavities [10,11]. In addition, six conserved cysteines form three interlocking disulfide bonds and fold to form a tight and stable hydrophobic binding cavity, which increases the structural stability of OBPs to a certain extent [12,13]. The stable sequence structure of OBPs plays an important role in maintaining their function.

Vogt and Riddiford (1981) first identified pheromone-binding proteins (PBPs) in the antennae of male *Antheraea polyphemus* [14]. OBPs have been discovered through transcriptome and genome analyses, and many functions of insect OBPs have been revealed through electrophysiology, insect behavior analysis, and in vitro binding experiments. The main function of OBPs is to recognize odor molecules and transport them to ORs [15]. OBPs can also increase the sensitivity of insect olfactory systems, regulate mating behavior, and participate in the tasting process [16,17,18,19,20]. For example, the RNAi-mediated downregulation of OBP56h expression alters the biosynthesis of epidermal pheromones, including the synthesis of 5-tricosene sex pheromones, resulting in delayed mating latency in *Drosophila melanogaster* [17]. Recently, RNAi technology has been widely used in integrated pest management research. The RNAi technology is a highly conserved in vivo mechanism for inhibiting gene expression. Studies on silencing target genes have promoted the exploration of insect gene functions [21]. 

The OBP function of *O. communna* has not been reported, and the role of *OcomOBPs* in recognition of *A. artemisiifolia* volatiles by *O. communa* is still unclear. In this study, the full-length sequence of *OcomOBP7* was cloned, and its sequence characteristics and expression profiles were analyzed. After the prokaryotic expression and purification of the *OcomOBP7* protein, fluorescence competitive binding analysis, RNAi, and electroantennography (EAG) experiments were performed to clarify the olfactory recognition mechanism and function of *OcomOBP7*. This study provides a theoretical basis for explaining the molecular mechanism of host recognition by *O. communa* and lays a foundation for the better biological control of *A. artemisiifolia* by *O. communa* in the future.

## 2. Materials and Methods

### 2.1. Insect Source 

The test insects were collected from the Langfang Experimental Station of the Chinese Academy of Agricultural Sciences, Hebei Province, China. All leaf beetles were raised on fresh *A. artemisiifolia* plants. The feeding environment was T = 26 ± 1 °C, RH = 70 ± 10%, and L:D = 14:10 h.

### 2.2. Gene Cloning and Sequence Analysis 

Total RNA was isolated from male and female antennae of 2–3-day-old *O. communa* using TRIzol reagent (Invitrogen, Carlsbad, CA, USA). First-strand cDNA was synthesized from 1 μg of total RNA using reverse transcriptase (TransGen Biotech, Beijing, China). Based on a previous antennal transcriptome of *O. communa* [22], specific primers were designed using Primer 5.0 software (PREMIER Biosoft International) to amplify the open reading frame of the *OcomOBP7* gene; the primer sequences are shown in Table 1.

The PCR product was purified and ligated into the pEASY-T3 vector (TransGen Biotech, Beijing, China). The ligation products were then transformed into Trans-T1 chemosensory cells (TransGen Biotech, Beijing, China) and coated on an AMP-resistant plate to screen for positive colonies.

SignalP 5.0 (http://www.cbs.dtu.dk/services/SignalP/ accessed on 21 February 2022) was used to predict the signal peptide of *OcomOBP7*. The Expasy Compute pl/mw tool (https://web.expasy.org/compute_pi/ accessed on 21 February 2022) was used to predict the molecular weight and isoelectric point. NCBI BLASTX (https://www.ncbi.nlm.nih.gov) was used to search for sequences similar to *OcomOBP7* in the database. Homology analysis was performed using ClustalW (https://www.genome.jp/tools-bin/clustalw accessed on 22 February 2022) for several similar sequences. ESPript 3.0 (https://espript.ibcp.fr/ESPript/cgi-bin/ESPript.cgi accessed on 22 February 2022) was used to perform multiple sequence alignments. Finally, phylogenetic tree analysis was performed using MEGA 5.0. The names and accession numbers of the proteins are listed in Appendix A.

### 2.3. Quantitative Real-Time PCR Analysis 

Total RNA was isolated from different tissues (male antennae, female antennae, heads, thoraxes, wings, legs, testes, and ovaries) of 2–3-day-old *O. communa* using TRIzol reagent (Invitrogen, Waltham, MA, USA). First-strand cDNA was synthesized from 1 μg of total RNA using reverse transcriptase (TransGen Biotech, Beijing, China). The expression of *OcomOBP7* in different tissues was investigated using real-time quantitative PCR (RT-qPCR). RT-qPCR was conducted on an ABI 7500 Fast Detection System (Thermo Scientific, Waltham, MA, USA) using Hieff qPCR SYBR Green Master Mix (TransGen Biotech, Beijing, China). The RT-qPCR amplification reaction conditions were as follows: denaturation at 95 °C for 5 min and 40 cycles at 95 °C for 10 s and 60 °C for 30 s, followed by melting curve analysis with instrument default settings. 

### 2.4. Heterologous Expression and Purification of OcomOBP7 

Based on the full-length gene sequence of *OcomOBP7*, the signal peptide was removed, and primers with restriction sites were designed to amplify the target fragment. The target fragment and pET28a vector were double-digested with restriction enzymes *Bam*HI and *Hin*dIII (Yeasen Biotech, Shanghai, China), and the target fragment was ligated into the expression vector. The recombinant plasmid was transformed into TOP10-competent cells, and the correctly sequenced recombinant plasmid was transformed into BL21(DE3)-competent cells (Yeasen Biotech, Shanghai, China). When the culture OD_600_ reached 0.6, the expression of the *OcomOBP7* protein was induced using isopropyl-β-D-thiogalactopyranoside (IPTG) at a final concentration of 1 mM at 37 °C for 10 h. The culture was centrifuged at 8000 rpm for 30 min, and the pellet was suspended in 1 × PBS, sonicated 200 times, and centrifuged. Inclusion bodies were denatured using 6 M guanidine hydrochloride and renatured using the redox method. Proteins were purified by HisTrap HP Ni ion affinity chromatography using the Rapid Protein Purification System KTA™ Avant 25 (General Electric, Boston, MA, USA). SDS-PAGE was used to monitor protein expression and purification. Protein concentration was determined using the Solarbio BCA Protein Concentration Assay Kit (Solarbio Science&Technology, Beijing, China).

After SDS-PAGE of the purified *OcomOBP7* protein, the target gel was cut off and sent to the Beijing Protein Innovation Company for liquid chromatography–mass spectrometry (LC-MS/MS).

### 2.5. Fluorescence Binding Assay 

Based on previous research on the volatiles of *A. artemisiifolia* [23], 26 standard chemical odors were selected to verify the binding characteristics of *OcomOBP7* using a fluorescence binding assay (Table 2).

Fluorescence competition binding experiments were performed using an F-380 fluorescence spectrophotometer (Gangdong Technology, Tianjin, China), and the quartz cuvette was 1 cm wide. The excitation and emission slits were 10 nm, the sensitivity was 2 s, the emission wavelength range was 360–500 nm, the excitation wavelength was 337 nm, and the scanning speed was 1200 nm/min. The fluorescent probe, 1-NPN, was dissolved in a chromatographic methanol solution at a concentration of 1 mM. The target protein was diluted to 2 μM with 50 mM Tris-HCl (pH 7.4), and 1 mL of protein diluent was added to the cuvette. The 1-NPN fluorescent probe was added to the protein diluent to obtain a final concentration of 2–20 μM and was allowed to stand for 30 s after mixing. The maximum fluorescence value for each addition of 1-NPN was recorded, and three replicates were used. The 1-NPN fluorescent probe was added to the protein diluent at a final concentration of 2 μM, mixed well, and allowed to stand for 30 s. The maximum fluorescence intensity of 1-NPN was recorded. The odorant ligand was then added at a final concentration of 2–20 μM, and the maximum fluorescence value produced by the mixed solution was determined.

The dissociation constants Ki of *OcomOBP7* and the fluorescent ligand compounds were calculated using the Scatchard equation. The formula is as follows: Ki = [IC50]/(1 + [1-NPN]/K1-NPN), where [1-NPN] represents the concentration of unbound 1-NPN, and K1-NPN is the binding constant of the *OcomOBP7*/1-NPN complex. 

### 2.6. RNAi-Mediated Gene Silencing 

dsRNA templates were synthesized by PCR using primers containing the T7 promoter sequence as templates. dsRNA was synthesized in vitro using a T7 RNAi Transcription Kit (Ambion Inc., Waltham, MA, USA) according to the manufacturer’s instructions. dsRNA was dissolved in DEPC water and its concentration determined using a Nanophotometer P330 (Implen, Germany). The quality of RNA was determined using 1% gel electrophoresis. Female and male adults with an initial emergence of <12 h were injected. A Nanoliter 2000 microinjector (WPI, Sarasota, FL, USA) was used to inject 1 ng of dsRNA into the pronotum of beetles. dsEGFP was used as a control. After injection, the beetles were kept in Petri dishes and fed fresh *A. artemisiifolia* leaves daily.

### 2.7. Electrophysiological Recordings

Based on the fluorescence binding experimental results, α-pinene, ocimene, and myrcene were used for the electrophysiological experiments. The universal hydrophobic solvent n-hexane was tested for use in EAG experiments [24,25]; the antennae of *O. communa* did not have an EAG response to it. Therefore, the odorants in this experiment were dissolved in n-hexane at a final concentration of 10 μg/μL, with n-hexane alone serving as a blank control. Filter paper strips with 10 μL of odorant were placed in a Pasteur Pipette for EAG experiments. The head and end of one antenna of the *O. communa* adult were cut off with a blade 48 h after the injection of dsRNA. A glass electrode filled with KCl conductivity liquid was inserted into the incision of the head of the *O. communa* and then connected to the electroantennogram system. The antenna incision was connected to a glass electrode at the other end using a micromanipulator. Each odorant was stimulated for 0.2 s with an interval of more than 30 s. Each treatment consisted of more than 20 biological replicates. Finally, the response curve was recorded using the EAG2000 software. 

### 2.8. Statistical Analysis

All statistical tests were conducted using IBM SPSS Statistics for Windows version 25.0 (IBM). Different tissue expression levels of *OcomOBP7* were calculated using the comparative 2^−∆∆Ct^ method. Multiple groups of data were compared using one-way ANOVA. The EAG data were analyzed using two independent samples for nonparametric test analysis. Differences were considered statistically significant at *p* < 0.05. Images were obtained using OriginPro 9.1 software and GraphPad Prism (version 8.0).

## 3. Results

### 3.1. Clone and Sequence Analysis of OcomOBP7

Based on the antennal transcriptome data of *O. communa*, the *OcomOBP7* gene was cloned; the complete ORF was 420 bp, encoding 139 amino acids (Appendix A).

The deduced *OcomOBP7* protein contained a signal peptide of 19 amino acids. The predicted molecular weight of the mature protein was 16.5 kDa, and the theoretical PI was 8.74. According to multiple sequence alignment results, the protein sequence similarity between *OcomOBP7* and other Coleoptera insects was 44–54%. The deduced amino acid sequence had six conserved cysteines and belonged to the classical OBP family. The cysteine mode of *OcomOBP7* was C1-X_23-44_-C2-X_3_-C3-X_36-43_-C4-X_8-12_-C5-X_8_-C6 (Figure 1a).

We constructed a phylogenetic tree of the *OcomOBP7* protein and homologous sequences of other related species of Coleoptera. According to the phylogenetic tree, *CforOBP6* and *HaxyOBP7* were clustered together, and other OBP proteins were clustered. the *OcomOBP7* and *CbowOBP1* of *Colaphellus bowringi* clustered on the same branch, with a similarity of up to 54% (Figure 1b).

### 3.2. Expression Profiles of OcomOBP7

The expression patterns of *OcomOBP7* were analyzed via RT-qPCR using RNA extracted from female and male antennae, heads, thoraxes, wings, legs, testes, and ovaries. The results showed that *OcomOBP7* expression was higher in antennae than in other tissues (Figure 2). There was no significant difference in expression between male and female antennae, indicating that *OcomOBP7* may be involved in chemical communication.

### 3.3. Expression and Purification of OcomOBP7

At 37 °C, the recombinant protein *pET28/OcomOBP7* was successfully expressed in *E. coli* after 10 h of induction with 1 mM IPTG. SDS-PAGE showed that the recombinant protein *pET28/OcomOBP7* was expressed in the inclusion bodies. After denaturation and renaturation, the *OcomOBP7* fusion protein was purified using nickel affinity chromatography. The band size in the SDS-PAGE was approximately 17 kDa, which was close to the expected molecular weight of 16.5 kDa (Appendix A).

The concentration of purified protein was 1.92 mg/mL. Since the 6 × His tag of *pET28a* is very small and has little effect on the function of the protein, the purified protein was directly used for fluorescence competitive binding experiments.

The LC-MS/MS identification results for *OcomOBP7* were retrieved by database construction. The results showed that the purified protein sample was OBP7 from *O. communa* (Appendix A).

### 3.4. Ligand-Binding Characteristic of OcomOBP7

To study the binding ability of *OcomOBP7* to *A. artemisiifolia* volatiles, we first determined the binding constant (Kd) of the protein and probe 1-NPN. The binding curve and Scatchard equation showed that *OcomOBP7* had strong binding ability with 1-NPN, and Kd was 1.47 ± 0.20 μM (Figure 3a).

The binding ability of *OcomOBP7* to 26 *A. artemisiifolia* volatiles was determined using a fluorescence competitive binding assay. The IC_50_ and dissociation constant (Ki) values are listed in Table 2. The smaller the Ki value, the stronger the binding ability was. Among the 26 tested volatiles, 14 *A. artemisiifolia* volatiles bound to *OcomOBP7*, including 12 alkenes, 1 alkane, and 1 ester. The binding curves of these 14 volatiles with *OcomOBP7* are shown in Figure 3. *OcomOBP7* bound most alkenes and had the strongest binding affinity to trans-β-farnesene, with a Ki value of 0.48 ± 0.04 μM. *OcomOBP7* showed high binding capacity with ocimene (1.23 ± 0.11 μM), (R)-(+)-dipentene (2.22 ± 0.11 μM), (S)-(-)-limonene (2.55 ± 0.40 μM), DL-limonene (2.84 ± 0.20 μM), α-phellandrene (2.91 ± 0.42 μM), myrcene (3.97 ± 0.85 μM), Y-terpinene (4.70 ± 0.32 μM), sabinene (4.75 ± 0.14 μM), and α-pinene (6.55 ± 0.05 μM). *OcomOBP7* also strongly bound to n-octane (0.67 ± 0.12 μM). Moderate binding ability was observed with camphene (9.85 ± 0.66 μM) and β-pinene (10.88 ± 0.59 μM). Borneol acetate showed a weak binding ability to *OcomOBP7* (20.37 ± 0.07μM) (Figure 3b,c).

### 3.5. RNAi and EAG Analysis

To further verify the biological function of *OcomOBP7*, EAG experiments were performed after the RNAi treatment. The product size of the dsRNA synthesized in this study was 420 bp, which was consistent with the target gene and could be used for subsequent microinjection. dsRNA was injected, and the interference efficiency was detected via RT-qPCR after 48 h. Compared with the injection of dsEGFP, the expression of *OcomOBP7* decreased significantly in the antennae of leaf beetles injected with dsOBP7 (*F* = 38,362.45, *p* < 0.0001), and the interference efficiency reached 98.4% (Figure 4a).

According to the fluorescence competitive binding experiment results, we selected the *A. artemisiifolia* volatiles with strong binding ability to *OcomOBP7* for electroantennogram experiments to verify the antennal response of *O. communa* to volatiles after 48 h of RNAi. Because not all *A. artemisiifolia* volatiles could activate antennal responses to *O. communa*, we selected three volatiles for RNAi verification. The average EAG responses of *O. communa* antennae to α-pinene, ocimene, and myrcene in the dsEGFP control group were 0.36 ± 0.03, 0.19 ± 0.01, and 0.10 ± 0.01, respectively (Figure 4b). After the injection of dsRNA, the EAG response of *O. communa* antennae significantly decreased to 0.28 ± 0.02 and 0.14 ± 0.01 for α-pinene (*p* = 0.010) and ocimene (*p* = 0.016), respectively. The EAG response of myrcene slightly decreased to 0.09 ± 0.01; however, it was not significantly different to the dsEGFP control group (*p* = 0.217).

## 4. Discussion

Antennae are the main organs of insects that receive external information and harbor various olfactory genes and proteins. OBPs are essential for insects to recognize external odors; they play important roles in regulating insect behavior and the chemical communication mechanisms of insects [26,27]. Based on previous studies, 25 putative odorant-binding proteins were identified in the antennal transcriptome of *O. communa* [22]. In the present study, we cloned the full-length *OBP7* gene of *O. communa* and analyzed its sequence characteristics, expression profiles, and binding affinities. The amino acid sequence of *OcomOBP7* contains six conserved cysteine sequences and a classical structure that belongs to the classical OBP family [28,29,30].

Studies have shown that the expression of OBPs in different tissues may be related to different physiological functions [31]. For example, the expression levels of *CsupPBP1* and *CsupPBP2* in the male antennae of *Chilo suppressalis* were significantly higher than those in females, and the two PBPs are involved in recognizing pheromones by male insects [16]. The expression levels of *AlinOBP11* in the adult legs of *Adelphocoris lineolatus* were significantly higher than those in other tissues, indicating that *AlinOBP11* has important gustatory functions in *A. lineolatus* [32]. The expression level of *OcomOBP7* in the antennae was significantly higher than in other tissues, indicating that it plays an important role in olfaction. In addition, its expression level was not significantly different between the male and female antennae, suggesting *OcomOBP7* is involved in olfactory behavior, such as host location, in both sexes.

The structural characteristics of OBP determine its ability to bind odorous molecules [33,34]. Fluorescence competitive binding experiments are now considered a popular method for screening OBP ligands [35]. The results of the fluorescence binding experiments in this study showed that *OcomOBP7* had a broad ligand-binding affinity; it could bind 14 of the 26 candidate *A. artemisiifolia* volatiles, including 12 alkenes, n-octane, and bornyl acetate. Previous studies have shown that the main volatiles of *A. artemisiifolia* include (E)-β-farnesene, ocimene, limonene, α-pinene, and myrcene [23]. *OcomOBP7* showed strong binding with (E)-β-farnesene, ocimene, α-pinene, and myrcene, indicating that these compounds may play an important role in host plant localization. Among these, (E)-β-farnesene had the strongest binding with *OcomOBP7*. (E)-β-Farnesene is a sesquiterpene compound with a strong aroma and biological activity [36], which may be one of the reasons for its strong binding with *OcomOBP7*. We also detected *OcomOBP7* binding to three isomers of limonene ((+)-dipentene, (S)-(−)-limonene, and DL-limonene). The results showed that the binding abilities of the three isomers were almost the same, indicating that the binding capacities of OBP to the isomers were similar. However, the binding ability of *CpunOBP8* to the two isomers of hexyl acetate and ethyl caprylate was significantly different in *Conogethes punctiferalis*. Of the eight compounds with the same molecular formula, C_10_H_16_, *CpunOBP9* and *CpunABP* bound strongly to only two. These results indicate that OBPs have strict requirements for the molecular configuration of odorant ligands [30].

RNAi and EAG experiments were performed to confirm the biological functions of *OcomOBP7* in *O. communa*. According to previous experiment results (unpublished data), α-pinene, ocimene, and mycere could significantly attract the beetles. Thus, to elucidate the olfactory communication mechanism between *A. artemisiifolia* and *O. communa*, although *OcomOBP7* was a broad-spectrum OBP gene, we focused on the *OcomOBP7*-binding ability of α-pinene, ocimene, and mycere. In addition, a previous experiment showed that limonene isomers and trans-β-farnesene cannot activate the antennal response via the EAG method. So, we put our focus on α-pinene, ocimene, and mycere. This is similar to s study on Asian honeybees; three odors with high binding affinity to *AcerOBP6* did not cause a strong reaction to the antennae, and there was no significant change in the EAG assay [37]. The RT-qPCR results showed that compared with dsEGFP-injected beetles, the expression level of *OcomOBP7* in beetles injected with dsOBP7 significantly reduced, and the interference efficiency reached 98.4%. In addition, compared to dsEGFP-injected beetles, the antennal response of dsOBP7-injected *O. communa* to α-pinene and ocimene decreased significantly. These experimental results indicated that α-pinene and ocimene are odorant ligands that correspond to *OcomOBP7*. From the perspective of pest control, modulating the OBP gene using RNAi and compromising the olfactory system could represent a novel method for controlling insect pests in future research [38].

In this study, we cloned the *OcomOBP7* gene and screened 26 *A. artemisiifolia* volatiles for binding ability using a fluorescence competitive binding assay to verify the ligands of *OcomOBP7* in vitro. RNAi combined with EAG experiments demonstrated that *OcomOBP7* was involved in recognizing α-pinene and ocimene. This study highlights the importance of the *O. communa*–*A. artemisiifolia* chemical communication mechanisms, which will lay a theoretical foundation for the development of beneficial insect attractants and help control ragweed through biological control.

## 5. Conclusions

We cloned the *OcomOBP7* gene from *O. communa* and expressed and purified the resulting protein. This protein is a classical OBP that is specifically expressed in the antennae of *O. communa*. *OcomOBP7* had an extensive ability to bind to alkenes. Among them, it had strong binding with limonene, α-pinene, and myrcene, the main volatile components of *A. artemisiifolia*. RNAi combined with EAG experiments further verified that α-pinene and ocimene are odor ligands of *OcomOBP7*, and *OcomOBP7* participates in the recognition process. In conclusion, the binding of *OcomOBP7* to *A. artemisiifolia* volatiles is broad-spectrum, indicating that it plays an important role in host plant localization.

## Figures and Tables

**Figure 1 insects-14-00190-f001:**
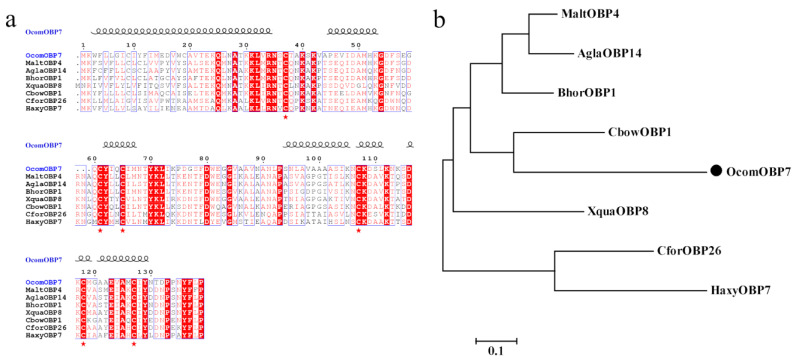
Sequence and phylogenetic analysis of *OcomOBP7*. (**a**) Amino acid sequence alignment of *OcomOBP7* with other homologous proteins. Target genes are marked in blue, conserved residues are highlighted in white letters with a red background, alignment positions are framed in blue box if the corresponding residues are identical or similar, and Six conserved cysteine residues are labeled with red stars, the helix represents the secondary structure of *OcomOBP7*. (**b**) Phylogenetic analysis of *OcomOBP7* and other homologous proteins. The target gene is marked with black dots.

**Figure 2 insects-14-00190-f002:**
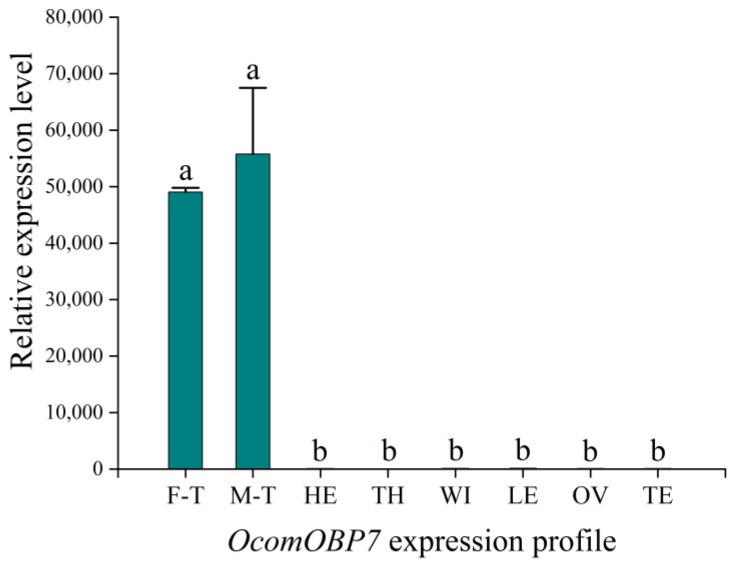
Expression profiles of *OcomOBP7*. F-T: female antennae; M-T: male antennae; HE: head; TH: thorax; WI: wing; LE: leg; OV: ovary; TE: testis. All values are shown as the mean ± SD. Different letters indicate significant differences at *p* < 0.05 by LSD test.

**Figure 3 insects-14-00190-f003:**
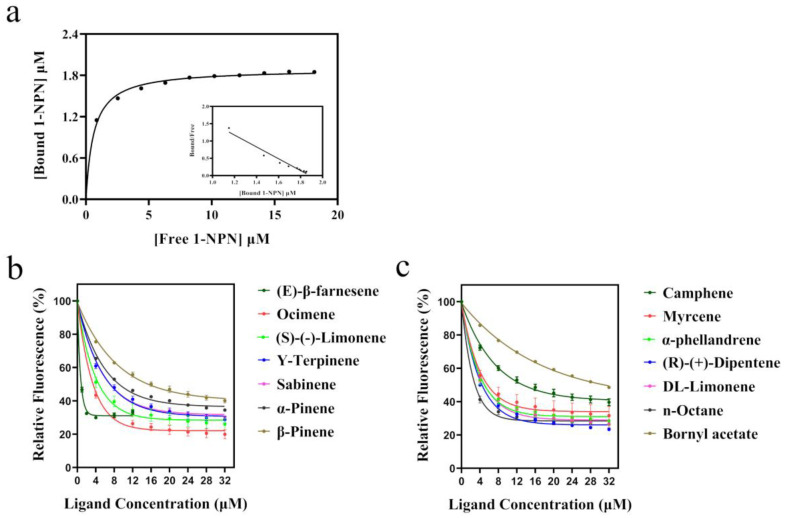
Ligand-binding experiment of *OcomOBP7*. (**a**) The binding curve of *OcomOBP7* and 1-NPN and Scathard equation. (**b**,**c**) The binding curve of *OcomOBP7* and *A. artemisiifolia* volatiles.

**Figure 4 insects-14-00190-f004:**
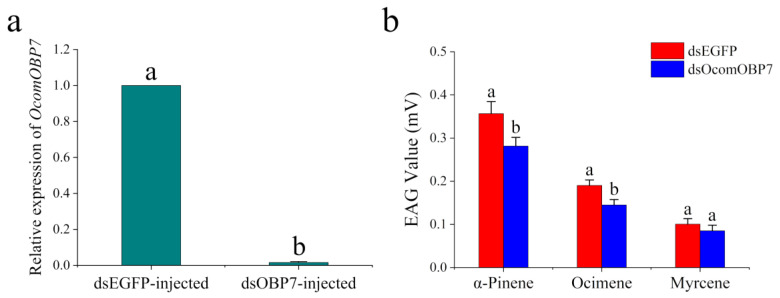
RNA interference of *OcomOBP7*. (**a**) The relative expression level of *OcomOBP7* after 48 h of RNAi. Different letters indicate significant differences at *p* < 0.05. (**b**) EAG response of *O. communa* to *A. artemisiifolia* volatiles 48 h after RNAi. The comparison between different treatment groups was analyzed using a nonparametric test. Different letters indicate significant differences at *p* < 0.05 using nonparametric test.

**Table 1 insects-14-00190-t001:** The primer sequence of *OcomOBP7* for cloning, RT-qPCR, and dsRNA.

Gene Name	Primer Name	Primer Sequence (5′-3′)
*OcomOBP7*	*OcomOBP7*-F	ATGAAGTGGTTCCTGCTT
	*OcomOBP7*-R	TTATGGAAGGAAATAATTTG
	*OcomOBP7*-F-q	GAAAAACAATTAAATGCGACCA
	*OcomOBP7*-R-q	ACCTTCCCAATCAAACGACC
	*OcomOBP7*-eF	GGATCCGCAGTTACAGAAAAACAAT
	*OcomOBP7*-eR	AAGCTTTTATGGAAGGAAATAATTTG
dsOcomOBP7	dsOcomOBP7-F	TAATACGACTCACTATAGGGATGGAAGATGTATGGTGTG
	dsOcomOBP7-R	TAATACGACTCACTATAGGGAAGGAAATAATTTGGAGG
dsEGFP	dsEGFP-F	TAATACGACTCACTATAGGGTGAGCAAGGGCGAGGAG
	dsEGFP-R	TAATACGACTCACTATAGGGCGGCGGTCACGAACTCCAG
*RL19*	*RL19*-F-q	AAGGAAGGCATTGTGGAT
	*RL19*-R-q	GACGCAAATCTCGCATAC

**Table 2 insects-14-00190-t002:** Standard chemical odors used in this study.

	CAS Number	Name of Compound	IC_50_ (μM)	Ki (μM)
Alkenes	18794-84-8	(*E*)-β-farnesene	0.76	0.48 ± 0.04
	13877-91-3	Ocimene	1.94	1.23 ± 0.11
	5989-54-8	(S)-(-)-Limonene	4.02	2.55 ± 0.40
	99-85-4	Y-Terpinene	7.42	4.70 ± 0.32
	3387-41-5	Sabinene	7.49	4.75 ± 0.14
	80-56-8	α-Pinene	10.34	6.55 ± 0.05
	2437-95-8	β-Pinene	17.18	10.88 ± 0.59
	87-44-5	β-Caryophyllene	-	-
	6753-98-6	α-Humulene	-	-
	565-00-4	Camphene	15.54	9.85 ± 0.66
	123-35-3	Myrcene	6.26	3.97 ± 0.85
	99-83-2	α-phellandrene	4.59	2.91 ± 0.42
	5989-27-5	(R)-(+)-Dipentene	3.50	2.22 ± 0.11
	124-76-5	Isoborneol	-	-
	464-49-3	Camphor	-	-
	138-86-3	DL-Limonene	4.48	2.84 ± 0.20
Alkanes	112-40-3	n-Dodecane	-	-
	629-50-5	n-Tridecane	-	-
	111-65-9	n-Octane	1.05	0.67 ± 0.12
Esters	3681-71-8	cis-3-Hexenyl Acetate	-	-
	125-12-2	Isobornyl acetate	-	-
	76-49-3	Bornyl acetate	32.16	20.37 ± 0.07
Aldehydes	124-13-0	Octanal	-	-
	124-19-6	1-Nonanal	-	-
Alcohols	78-70-6	Linalool	-	-
	18479-58-8	Dihydromyrcenol	-	-

## Data Availability

Data are contained within the article or Appendix A.

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
