# Peer review of "Characterization and Functional Analysis of *OcomOBP7* in *Ophraella communa* Lesage"

_insects, 2023, doi:10.3390/insects14020190_

Round 1
Reviewer 1 Report
The abstract and discussion part needs to be extensively improved. The study is useful, but authors need to highlight the significance of research particularly how these results will contribute to understanding of Ophraella communa olfaction and controlling the Ambrosia artemisiifolia. Authors also need to explain why they have selected specific OBP for fucntional characterization.
Author Response
请参阅附件。

Reviewer 2 Report
This manuscript by Sue et al details isolation and functional characterization of an OBP from an oligophagous beetle which is currently used to control the invasive A. artemisiifolia in China.
Overall the study is sound, but I do have some questions:
1. the authors should explain better why n-hexane instead of another hydrophobic solvent solvent such as isopropyl myristate is used in their experiments
2. line 259: the authors refer to A. artemisiifolia volatiles, but they do not mention where this information comes from (ie reference), or which these volatiles are. Are they all the ones tested in Fig 5? Are the ones tested in Fig 5 a subset of the total volatiles? Are the ones tested in the list of such volatiles at all?
3. Regarding Fig 6.
a. Should't the authors have tried ds against the closest OBP to OBP7 to demonstrate specificity of their dsRNA approach?
b. I am perplexed as to why the authors report the effect of OBP7 abrogation on the odorants presented on Fig when they report that the strongest binder to OBP7 is in fact trans-β-farnesene, easily acquired limonene isomers and others and focus on α-pinene which is not the best candidate in my understanding of the data?
Finally a few corrections:
line 42: replace "its" with "their"
table 2 and Fig5b: bata-pinene should be bEta pinene.
line 181: what does singleness mean??
line234: .........may be involved in chemical communication (remove: "the" and "function" from the sentence)
Reviewer 3 Report
In this study, the authors cloned the full-length sequence of OcomOBP7 and analyzed the sequence characteristics and expression profiles. Besides, the fluorescence competitive binding analysis, RNAi, and EAG experiments were carried out to clarify the olfactory recognition mechanism and function of OcomOBP7. This study provides a theoretical basis for explaining the molecular mechanism of host recognition by O. communa and lays a theoretical foundation for better biological control of A. artemisiifolia by O. communa in the future. In general, proper scientific methods were used, and the results are of interest. However, there are some points for further improvement of the manuscript which should be addressed before possible publication.
-The manuscript needs careful proofreading and revision. Grammar mistakes are undermining the significance of this study. I strongly encourage authors to polish this manuscript by an English editing company or through a native speaker.
-The authors should add some detailed results. Or at least they should rewrite the current results in little bit detail.
-Starting the introduction section with insects is too general. The authors should revise the introduction section especially the starting paragraph. This should be more specific and to the point. Try to focus on target insects, and avoid the general sentences.
-In introduction section, the authors should add more detail about the RNAi technology and their use in insect pest control. The following key documents can be useful for this manuscript.
- List, F., Tarone, A. M., Zhu‐Salzman, K., & Vargo, E. L. (2022). RNA meets toxicology: efficacy indicators from the experimental design of RNAi studies for insect pest management. Pest Management Science. https://doi.org/10.1002/ps.6884
- Ullah et al. 2022. RNA interference-mediated silencing of ecdysone receptor (EcR) gene causes lethal and sublethal effects on melon aphid, Aphis gossypii. Entomologia Generalis, 42 (5), 791-797.
- Section 2.6. RNAi-mediated Gene Silencing: What is the product size of dsRNA? Please add in main text as I suggest authors to shift Figure 1 to supplementary file.
- In results, please add complete statistical values. The exact values…
- I suggest authors to shift Figure 4 to supplementary file.
- The quality of Figure 5 is not okay for a scientific publication. The authors should increase the font size of legends, and have minimum 300 dpi resolution. Please increase the figures size.
- The discussion section is too short and general. The authors need to work more on this section. They need to justify and compare their main findings in more detail with recently published work specifically focusing on target insects.
- Some references are too old, please replace them with latest work, from quality journals.
Round 2
Reviewer 2 Report
Dear authors,
thanks for your replies to my enquiries. I do believe however that the contents of this paragraph must be incorporated in the manuscript because it explains use and focus on these specific volatiles. You may just reference as data not shown the previous experiment results (unpublished data), α-pinene, ocimene and mycere could significantly attract the beetles,
or include that data as supplemental. The same for the data below]
previous experiment showed that limonene isomers and trans-β-farnesene can not activate the antennal response by EAG method
Dear editor and reviewer, according to previous experiment results (unpublished data), α-pinene, ocimene and mycere could significantly attract the beetles. Thus to elucidate the olfactory communication mechanism between A. artemisiifolia and O. communa, although OcomOBP7 was a broad-spectrum OBP gene, we focus on the OcomOBP7 binding ability of α-pinene, ocimene and mycere. In addition, previous experiment showed that limonene isomers and trans-β-farnesene can not activate the antennal response by EAG method. So we put our focus on α-pinene, ocimene and mycere.
Author Response
Response:Line343-349: Thanks for your question. I have added this paragraph into the manuscript according to your suggestion.
Reviewer 3 Report
No further comments. I think this revised version can be accepted for publication in insects
Author Response
Response: Dear editor and reviewer, I have checked and corrected the English language and spelling. Thanks again for your advice.